# ThinkOmni: Lifting Textual Reasoning to Omni-modal Scenarios via Guidance Decoding

**Yiran Guan[1]**    **Sifan Tu[1]**    **Dingkang Liang[1]**    **Linghao Zhu[1]**
**Jianzhong Ju[2]**    **Zhenbo Luo[2]**    **Jian Luan[2]**    **Yuliang Liu[1]**    **Xiang Bai[1]✉**
[1]Huazhong University of Science and Technology    [2]MiLM Plus, Xiaomi Inc.
{yiranguan, dkliang, xbai}@hust.edu.cn

## Abstract

Omni-modal reasoning is essential for intelligent systems to understand and draw inferences from diverse data sources. While existing Omni-modal Large Language Models (OLLM) excel at perceiving diverse modalities, they lack the complex reasoning abilities of recent Large Reasoning Models (LRM). However, enhancing the reasoning ability of OLLMs through additional training presents significant challenges, including the need for high-quality data, task-specific adaptation, and substantial computational costs. To address these limitations, we propose **ThinkOmni**, a training-free framework that lifts textual reasoning to omni-modal scenarios. ThinkOmni introduces two key components: 1) *LRM-as-a-Guide*, which leverages off-the-shelf LRMs to guide the OLLM decoding process; 2) *Stepwise Contrastive Scaling*, which adaptively balances perception and reasoning signals without manual hyperparameter tuning. Experiments on six multi-modality reasoning benchmarks demonstrate that ThinkOmni consistently delivers performance improvements, with main results achieving $70.2\%$ on MathVista and $75.5\%$ on MMAU. Overall, ThinkOmni offers a flexible and generalizable solution for omni-modal reasoning and provides new insights into the generalization and application of reasoning capabilities. Project page: **https://1ranguan.github.io/thinkomni**

## 1    Introduction

The emergence of Large Reasoning Models (LRMs) marks a paradigm shift from the traditional *fast thinking* of standard LLMs, which rely on immediate intuition, to *slow thinking*, which emphasizes reflective and iterative reasoning. Recent LRMs, such as DeepSeek-R1 (Guo et al., 2025) and o1 (OpenAI, 2025), have demonstrated exceptional performance in specialized reasoning tasks like mathematical problem-solving and code generation. Nonetheless, their effectiveness remains predominantly constrained to textual inputs, thus limiting their applicability to more complex, omni-modal real-world scenarios involving text, audio, images, and videos (see Fig. 1(a)).

Omni-modal reasoning is essential for synthesizing diverse data sources and enabling sophisticated inference in context-rich tasks. Strong omni-modal reasoning capabilities have profound implications for practical applications such as advanced virtual assistants (Zhang et al., 2025) and embodied robots (Gan et al., 2020). Although recent advances in Omni-modal Large Language Models (OLLM) (Xu et al., 2025; Li et al., 2025b; Liu et al., 2025c; Fu et al., 2025; Luo et al., 2025) have shown promise in comprehending various input modalities, these models typically fall short when tasked with intricate reasoning across modalities, as illustrated in Fig. 1(b). Therefore, a fundamental research challenge is how to effectively extend and elevate the reasoning capabilities of models from primarily textual inputs to truly omni-modal scenarios.

Actually, this is not a trivial problem, and despite considerable efforts, existing approaches to omni-modal reasoning are still limited in several critical aspects. Specifically, 1) Insufficient modality diversity. Current studies largely focus on specific modalities (e.g., image (Liu et al., 2025b;a; Lin et al., 2025), audio (Li et al., 2025a), or video (Wang et al., 2025)), rather than generalizing across

---

Work done at Xiaomi Inc.    ✉ Corresponding author.

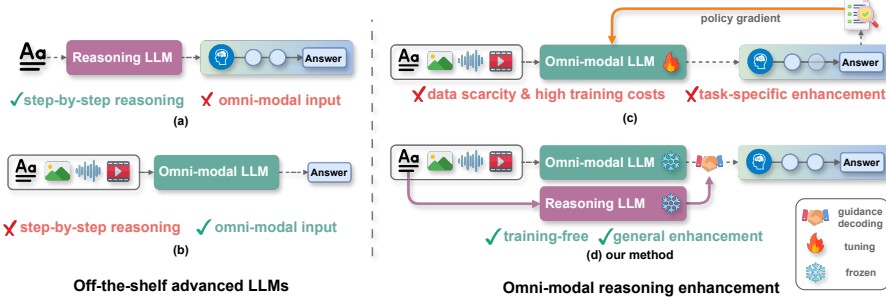

Figure 1: **Comparison between existing methods and THINKOMNI.** We integrate an OLLM with an LRM via guidance decoding, enabling advanced reasoning abilities with omni-modal input.

arbitrary combinations of modalities. 2) Task-specific enhancement. Enhancements proposed for existing OLLMs (Zhao et al., 2025; Zhong et al., 2025; Rouditchenko et al., 2025; Yang et al., 2025b) remain confined to particular downstream tasks, lacking broader generalizability. 3) Data scarcity and high training costs. Current methods predominantly rely on extensive supervised finetuning (SFT) (Xu et al., 2024; Yang et al., 2025c) (requiring tens of thousands of reasoning examples) or reinforcement finetuning (RFT) (Shao et al., 2024; Yu et al., 2025) approaches demanding training computational resources (e.g., $8{\times}40$G VRAM for 7B model, $16{\times}80$G VRAM for 32B model). These challenges collectively motivate an important question: *Is it possible to overcome the constraints of data and training conditions to bring general reasoning abilities to omni-modal content?*

In this paper, we propose THINKOMNI, a novel training-free framework designed to lift textual reasoning to omni-modal scenarios (see Fig. 1(d)). Unlike existing approaches (see Fig. 1(c)) reliant on costly data annotation or additional model training, THINKOMNI directly leverages off-the-shelf LRMs as decoding-time guides for OLLMs. Specifically, we first introduce the *LRM-as-a-Guide* strategy, enabling the integration of reasoning capabilities from LRMs into OLLMs. We further identify a potential issue: a fixed guidance weight is unsuitable for all the tasks, and manual, task-specific adjustment is impractical. To resolve this, we propose a *Stepwise Contrastive Scaling* module, adaptively balancing perceptual and reasoning signals based on real-time analysis of model predictions. This module adapts to various task types and facilitates coherent omni-modal reasoning.

Extensive experiments conducted on six challenging multi-modal reasoning benchmarks demonstrate the effectiveness of our method. Specifically, our method improves the state-of-the-art open source OLLM Qwen2.5-Omni (Xu et al., 2025) by substantial margins without additional training, as shown in Fig. 2, rivaling or surpassing models that undergo extensive RFT. Additionally, compared to other guidance decoding algorithms (Li et al., 2022; Liu et al., 2024), our method reduces the burden of multi-modal data input, thereby maintaining decoding efficiency.

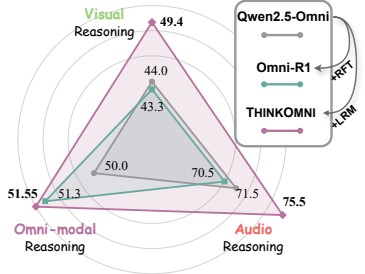

Figure 2: Performance comparison.

THINKOMNI provides a flexible framework for lifting textual reasoning to a more diverse and enriched input space. By leveraging the strengths of OLLM and LRM, we explore the effective generalization of reasoning capabilities to omni-modal scenarios in a training-free manner. Besides, our method is not limited to current LRMs. As new LLM technologies emerge (often developing faster than multi-modal variants), our approach can be easily adapted to improve performance across multi-modal variants and other downstream domains.

## 2 PRELIMINARIES

### 2.1 NEXT TOKEN PREDICTION

Given an omni-modal input $O$ (e.g., images, audios, videos) and a sequence of text tokens $x_{<t} = (x_1, x_2, \ldots, x_{t-1})$, the OLLM $M$ first computes the logits $z_t$ for the next token $x_t$ as

$z_t = M(x_{<t}, O)$, where $z_t \in \mathbb{R}^V$ and $V$ is the vocabulary size. The probability distribution $P$ for $x_t$ is then given by

$$P(x_t \mid x_{<t}, O) = \text{Softmax}(z_t). \tag{1}$$

Then $x_{<t+1} = (x_1, x_2, \ldots, x_t)$. The model computes a distribution and decodes a token at each step, resulting in an auto-regressive generation process.

## 2.2 Inference-time Guidance Decoding

Finetuning large language models is time-consuming and costly, highlighting the need for methods to modify or control models' behaviors without additional training. In this subsection, we introduce the following works to understand our method better: Contrastive Decoding (Li et al., 2022), Visual Contrastive Decoding (Leng et al., 2024), ProxyTuning (Liu et al., 2024), and ProxyThinker (Xiao et al., 2026). For models within the same family (i.e., sharing the same token vocabulary), these methods guide base model decoding by introducing a contrastive pair at the logits level:

$$\hat{z} = z^{\text{base}} + \alpha \cdot \underbrace{(z^+ - z^-)}_{\text{contrastive pair}}, \tag{2}$$

where $\alpha$ controls the influence of the guidance signal. Here, $z^+$ and $z^-$ represent the logits from the positive and negative references, respectively. These encourage or discourage certain behaviors in the model's output. This mechanism is analogous to a differential amplifier circuit, which amplifies the desired signals while suppressing noise. Consequently, the model can reduce hallucinations or achieve preference alignment during inference without additional training.

In Contrastive Decoding (Fig. 3(a)), the contrastive pair is formed by comparing the responses to the same prompt from the original guiding model and an additional amateur model, with $z^+$ set to $z^{\text{base}}$. In Visual Contrastive Decoding (Fig. 3(b)), the contrastive pair is created by applying different input conditions to the same model. Specifically, $z^-$ is obtained by adding Gaussian noise to the input image and then performing inference. In contrast to these approaches, ProxyTuning and ProxyThinker (Fig. 3(c))

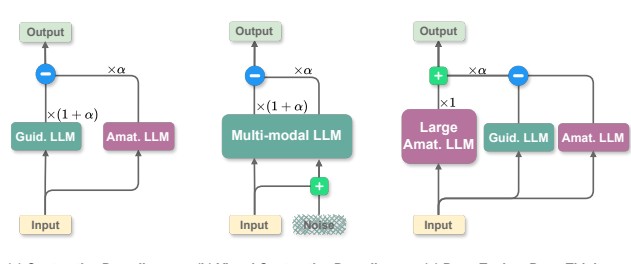

(a) Contrastive Decoding   (b) Visual Contrastive Decoding   (c) ProxyTuning, ProxyThinker

Figure 3: **Guidance decoding methods.** "Guid." denotes the guiding model, and "Amat." denotes the amateur model.

construct contrastive pairs across different models within the same family, aiming to transfer behaviors from more minor, guiding models to larger, amateur models.

Existing guidance decoding methods are limited to scenarios with **consistent input modalities** and **available expert models**. There is often no suitable expert model in omni-modal settings or other downstream tasks, making it technically challenging to construct effective guidance signals. Moreover, the heterogeneity of modalities complicates the alignment and integration of guidance during inference. Our work addresses these challenges by designing a framework for cross-modal guidance decoding, enabling preference alignment without requiring modality-specific expert models.

## 3 Method

This section outlines the implementation roadmap of THINKOMNI, starting with a straightforward guidance decoding approach. We first introduce *LRM-as-a-Guide*, which separates the input modalities of the Omni-modal Large Language Model (OLLM) and incorporates an off-the-shelf Large Reasoning Model (LRM) as a guiding component. While this approach is practical, coordinating fixed guidance decoding hyperparameters remains challenging due to the varying demands for reasoning signals across different tasks and scenarios. To address this shortcoming, we propose *Stepwise Contrastive Scaling*, a module that dynamically adjusts parameters based on real-time analysis of model predictions, thereby adapting automatically to each decoding scenario. An overview of our framework is provided in Fig. 4.

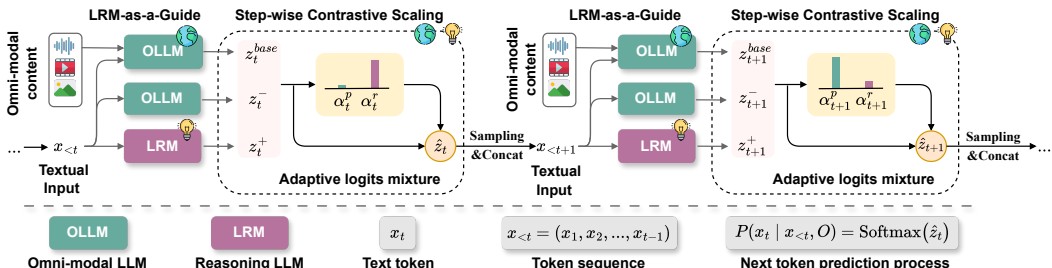

Figure 4: **Overview of THINKOMNI.** The framework begins by separating input modalities of the OLLM and introducing the LRM as a guiding model. Stepwise Contrastive Scaling dynamically adjusts guidance parameters based on real-time prediction analysis, enabling adaptive and effective decoding across diverse tasks.

## 3.1 LRM-AS-A-GUIDE

As discussed in Sec. 2.2, to address the gap where current guidance decoding approaches are limited to models with matched input modalities, we introduce the **LRM-as-a-Guide**, which lifts advanced textual reasoning into the omni-modal content through collaborative decoding with OLLM.

Let $M_O$ denote the OLLM and $M_R$ denote the LRM. As shown in Fig. 6(a), we compute the base logits with full omni-modal input, $z^{\text{base}} = M_O(x_{<t}, O)$. Then we discard the omni-modal content and feed $M_O$ only the textual prefix $x_{<t}$. The results are treated as the negative logits $z^- = M_O(x_{<t})$. The positive logits are produced by the LRM on the same prefix $z^+ = M_R(x_{<t})$. As formulated in Eq. (2), the token probability distribution will then serve as

$$\hat{P} = \text{Softmax}\Big[ M_O(x_{<t}, O) + \alpha \cdot \big(M_R(x_{<t}) - M_O(x_{<t})\big) \Big], \tag{3}$$

where the scalar $\alpha$ determines the extent to which the LRM influences the OLLM. After obtaining the mixed logits, we normalize them to probabilities and then sample the next token as usual.

Although the LRM cannot access omni-modal information, we mitigate this disadvantage and amplify the reasoning preference through the logits contrastive. During the generation process, the OLLM and LRM collaborate in a complementary manner. The OLLM, serving as the primary agent, extracts and integrates omni-modal clues, while the LRM provides deeper reasoning over the textual trace. As decoding progresses, the LRM can compensate for the lack of omni-modal information by leveraging the already decoded tokens, and the OLLM achieves logical reasoning through the reasoning preferences supplied by the LRM. Their strengths are seamlessly fused through logit mixing, resulting in a unified decoding framework that effectively integrates perception and reasoning.

## 3.2 STEPWISE CONTRASTIVE SCALING

While LRM-as-a-Guide effectively enables collaboration between the LRM and OLLM, there remains room for improvement regarding the choice of the fixed guidance weight $\alpha$. A fixed $\alpha$ may not consistently achieve the optimal balance between perception and reasoning across different tasks.

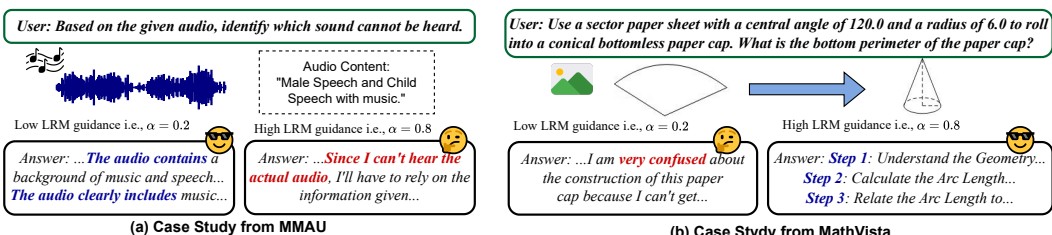

Figure 5: **Case studies from (a) MMAU and (b) MathVista.** (Sakshi et al., 2025; Lu et al., 2024) Tasks require different levels of LRM involvement. Using a fixed $\alpha$ limits the ability of the model to optimally adapt to task-specific needs, highlighting the need for a more flexible approach.

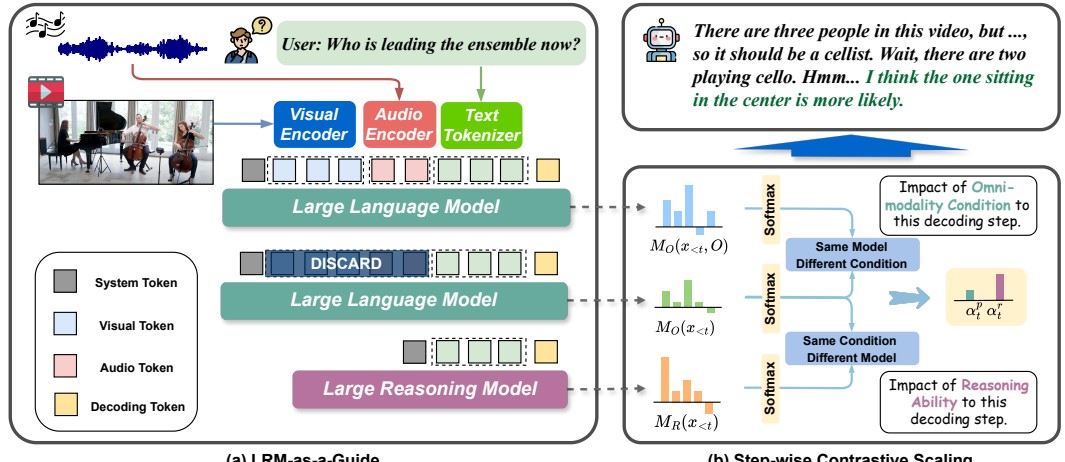

Figure 6: **Detailed process of THINKOMNI.** (a) The OLLM handles multi-modal inputs, while the LRM focuses on textual reasoning. By mixing their logits, the system effectively integrates perception and reasoning during token generation. (b) Each decoding step balances perception and reasoning dynamically by comparing logit distributions under different conditions and models.

The OLLM prefers a smaller $\alpha$ to emphasize omni-modal cues, while the LRM benefits from a larger $\alpha$ to strengthen its guidance. Additionally, since $z^+$ and $z^-$ do not possess comprehensive omni-modal content, excessive reliance on them (adopting a large $\alpha$) can lead to recognition bias such as hallucination (Fig. 5(a)). Conversely, setting $\alpha$ too low may diminish the effectiveness of guidance, thereby constraining the logical reasoning capabilities (see Fig. 5(b)). Motivated by this, we propose **Stepwise Contrastive Scaling**, which dynamically apportions a token's prediction budget between perception and reasoning through online analysis of logits.

We introduce a stepwise influence metric to determine whether each decoding step is dominated by perception or reasoning. Specifically, all the generated logits are first transformed into probability distributions with a softmax function, let $P_O$, $P_R$, $P$ denote the corresponding distributions for $M_O(x_{<t}, O)$, $M_R(x_{<t})$, and $M_O(x_{<t})$, respectively. The pairwise distances between these distributions are then quantified by the Jensen–Shannon divergence, which is employed in DoLa (Chuang et al., 2024) to measure the disagreement between two logits. This metric is symmetric, bounded, and numerically stable, making it well-suited for our purposes:

$$D_R = \mathrm{JS}\big(P_R \parallel P\big), \qquad D_P = \mathrm{JS}\big(P_O \parallel P\big). \tag{4}$$

Intuitively, $D_R$ reflects the unique influence of reasoning preference, whereas $D_P$ captures the contribution from perceptual omni-modalities. A larger pairwise distance signifies that the corresponding factor (perception or reasoning) impacts the current decoding step more. Building on this metric, we proceed to reformulate Eq. (3) and introduce an additional contrastive logits term:

$$\hat{P} = \mathrm{Softmax}\Big[ M_O(x_{<t}, O) + \alpha_t^r \cdot \big(M_R(x_{<t}) - M_O(x_{<t})\big) + \alpha_t^p \cdot \big(M_O(x_{<t}, O) - M_O(x_{<t})\big)\Big], \tag{5}$$

where $\alpha_t^r$ acts as the original guidance weight, capturing enhanced reasoning capability, whereas the difference contributed by $\alpha_t^p$ serves as an aggressive visual contrastive term (Leng et al., 2024) (i.e., by directly removing non-textual inputs rather than adding noise), reflecting augmented perceptual capability. To improve decoding stability, we employ a normalization strategy to ensure $\alpha_t^r + \alpha_t^p = 1$, with the coefficients determined by the relative magnitudes of $D_R$ and $D_P$. During the initial decoding steps, we constrain the magnitude of $\alpha_r$ to implement a warmup for the reasoning task. More implementation details are provided in Appendix B.

As shown in Fig. 4, the entire THINKOMNI procedure is training-free, requiring no additional finetuning or corpus statistics. Leveraging stepwise contrastive scaling, LRM-as-a-Guide can autonomously evaluate the relative contributions of perceptual and reasoning signals at each generation step, seamlessly balancing these complementary abilities without manual hyperparameter tuning.

Table 1: **Model performance on several omni-modal reasoning benchmarks.** Here, DeepSeek refers to DeepSeek-R1-Distill-Qwen-7B (Guo et al., 2025), and Qwen3 denotes Qwen3-8B (Yang et al., 2025a). The numbers in parentheses indicate the performance changes compared to the base OLLMs Qwen2.5-Omni-3B / 7B (Xu et al., 2025) and Omni-R1 (Zhong et al., 2025). Models marked with '*' are evaluated using our own evaluation scripts.

| Model | MathVista test-mini | MathVision test | MathVerse test-mini | MMAU(v05.15.25) test-mini | DailyOmni test | OmniBench test |
|---|---|---|---|---|---|---|
| *Close-Sourse Models* | | | | | | |
| GPT-4o | 63.8 | 30.4 | 50.8 | 62.5 | 56.5 | - |
| Gemini-2.0-Flash | 73.1 | 41.3 | 59.3 | 70.5 | 67.8 | - |
| *Open-Sourse Omni Models* | | | | | | |
| Baichuan-Omni-1.5 | 63.6 | - | - | 66.2 | 50.0 | 42.9 |
| Ola | 68.4 | - | - | 70.3 | 50.71 | - |
| *Open-Sourse RFT Omni Models* | | | | | | |
| Omni-R1* | 64.7 | 25.4 | 39.8 | 70.5 | 59.6 | 43.0 |
| HumanOmniV2* | 68.8 | 25.4 | 37.3 | 75.3 | 58.5 | 41.9 |
| *THINKOMNI-Qwen2.5-Omni-3B* | | | | | | |
| Qwen2.5-Omni-3B | 56.0 | 18.2 | 32.0 | 69.4 | 56.6 | 37.5 |
| + DeepSeek | 56.1(+0.1) | 20.2(+2.0) | 33.5(+1.5) | 70.1(+0.7) | 57.1(+0.5) | 39.9(+2.4) |
| + Qwen3 | 58.1(+2.1) | 25.3(+7.1) | 38.8(+6.8) | 70.6(+1.2) | 57.3(+0.7) | 39.5(+2.0) |
| *THINKOMNI-Qwen2.5-Omni-7B* | | | | | | |
| Qwen2.5-Omni-7B | 66.8 | 25.0 | 40.2 | 71.5 | 57.9 | 42.1 |
| + DeepSeek | 68.8(+2.0) | 28.2(+3.2) | 42.0(+1.8) | 73.8(+2.3) | 59.8(+1.9) | 43.2(+1.1) |
| + Qwen3 | 70.2(+3.4) | 32.9(+7.9) | 45.1(+4.9) | 75.5(+4.0) | 59.5(+1.6) | 43.6(+1.5) |
| *THINKOMNI-Omni-R1-7B* | | | | | | |
| Omni-R1 | 64.7 | 25.4 | 39.8 | 70.5 | 59.6 | 43.0 |
| + DeepSeek | 66.1(+1.4) | 27.0(+1.6) | 43.1(+3.3) | 73.1(+2.6) | 60.3(+0.7) | 43.5(+0.5) |
| + Qwen3 | 71.3(+6.6) | 31.5(+6.1) | 45.2(+5.4) | 75.4(+4.9) | 59.8(+0.2) | 43.4(+0.4) |

## 4 EXPERIMENT

### 4.1 EXPERIMENT SETUP

**Models** To validate the effectiveness of THINKOMNI, we conduct experiments on three OLLMs: Qwen2.5-Omni-3B / 7B (Xu et al., 2025) and Omni-R1 (Zhong et al., 2025). We utilize the DeepSeek-R1-Distill series (Guo et al., 2025) and the Qwen3 series (Yang et al., 2025a), both in thinking mode, as our LRMs to guide decoding.

**Benchmarks** To demonstrate the generalizability of THINKOMNI, we evaluate it on omni-modal scenarios using six benchmarks, comprising over $10,000$ test samples in total: MathVista (test-mini) (Lu et al., 2024), MathVision (Wang et al., 2024), MathVerse (test-mini) (Zhang et al., 2024), MMAU-v05.15.25 (test-mini) (Sakshi et al., 2025), Daily-Omni (Zhou et al., 2025), OmniBench (Li et al., 2024). More details are provided in Appendix B.

**Evaluation** We first use template matching for multiple-choice questions to extract the option from the model's output. If the answer cannot be extracted directly, we use GPT-4o to extract it and then compare the extracted answer to the gold answer. For free-form questions, we first use GPT-4o to extract the answer from the model's output, then compare the extracted answer to the gold answer to determine if their meanings are consistent. This process is designed to account for various expressions in the answers.

### 4.2 MAIN RESULT

To evaluate the generality and scalability of THINKOMNI, we benchmark the improvements of different LRM guides on several OLLMs with varying capability levels. Our main results are presented in Tab. 1. The experiment result shows that THINKOMNI brings extensive improvements across all OLLMs, LRMs, and benchmarks. For example, with the Qwen3 guide, THINKOMNI brings remarkable improvement to Qwen2.5-Omni-7B on MathVision by 7.9%, achieving the final score of 32.9%. Since LRMs do not have access to omni-modal data contents, our results demonstrate that THINKOMNI indeed lifts the complex reasoning of LRMs to the omni-modal scenario.

Table 2: **Comparison with several training-free methods.** All are built upon Qwen2.5-Omni-7B.

| Method | MathVista
test-mini | MMAU(v05.15.25)
test-mini | OmniBench
test |
|---|---|---|---|
| Base Model | 66.8 | 71.5 | 42.1 |
| Average Logits Fusion | 55.0(-11.8) | 55.7(-15.8) | 36.1(-6.0) |
| Caption-then-Answer | 61.0(-5.8) | 59.7(-11.8) | 32.3(-9.8) |
| VCD | 66.5(-0.3) | 72.2(+0.7) | 43.1(+1.0) |
| THINKOMNI (Ours) | 68.8(+2.0) | 73.8(+2.3) | 43.2(+1.1) |

We compare our approach with methods trained using reinforcement learning finetuning (RFT) (i.e., Omni-R1 (Zhong et al., 2025) and HumanOmniV2 (Yang et al., 2025b)). Based on the same foundation model, Qwen2.5-Omni-7B, our DeepSeek-guided model achieves comparable performance, while our Qwen3-guided model consistently outperforms all the RFT-based methods. Moreover, our approach can be applied to models already undergoing RFT, further demonstrating broad performance improvements.

In addition, we observe differences in performance gains, which the following factors can explain: 1) the capabilities of LRM models, newer models like Qwen3, with stronger logical understanding and reasoning abilities, achieve greater improvements compared to DeepSeek under identical settings; 2) the training data of LRMs is biased towards scientific and mathematical content, leading to more pronounced gains on these tasks; 3) the tested tasks themselves differ in their demands for reasoning ability, with scientific and mathematical tasks typically requiring more reasoning than audio or general omni-modal tasks.

### 4.3 COMPARE WITH TRAINING-FREE METHODS

We use the original evaluation results of the OLLMs as a baseline. In addition, we compare our method with several other training-free methods: 1) *Average Logits Fusion*, which directly averages the output logits of the OLLM and LRM during inference. 2) *Caption-then-Answer*, where the OLLM generates a detailed caption for the omni-modal input, and the LRM answers the question based on this caption. 3) *Visual Contrastive Decoding* (VCD) (Leng et al., 2024), which mitigates hallucinations by contrasting predictions from original inputs with those from distorted ones. In this ablation, we implement the negative baseline by directly removing the multi-modal inputs. As shown in Tab. 2, THINKOMNI significantly outperforms the base OLLM. For *Average Logits Fusion*, although simple mixing of logits allows the model to generate outputs, it negatively impacts answer accuracy due to improper integration. The *Caption-then-Answer* experiment demonstrates that when the LRM alone is responsible for answering, even with multi-modal information provided by the OLLM, performance drops significantly because information transmission is one-way. The OLLM cannot respond to the LRM's specific needs. *VCD* is designed to enhance attention to multi-modal information rather than reasoning ability, so its performance declines on MathVista, which requires stronger reasoning skills.

### 4.4 ABLATION STUDY

**Ablation study on fixed $\alpha$ and adaptive $\alpha^r$**  OLLM has limited capability in complex reasoning, while LRM cannot access multi-modal content. Over-reliance on either component leads to suboptimal performance. As shown in Fig. 7(a), adjusting the fixed guiding weight $\alpha$ markedly impacts results: when $\alpha = 0$, performance matches the original OLLM, and extreme $\alpha$ values reduce scores on both benchmarks. In contrast, our Stepwise Contrastive Scaling (Full THINKOMNI) consistently achieves superior results across both benchmarks. Furthermore, Fig. 7(b) visualizes the distribution of the dynamic $\alpha^r$, revealing distinct shifts across different tasks and underscoring the adaptive nature of our method in autonomously tuning parameters to meet specific task requirements.

**Ablation study on different LRMs**  Fig. 8 demonstrates the performance of Qwen2.5-Omni-7B under the guidance of LRM across different model sizes and series. When the LRM size is too small, performance on MathVista and OmniBench both degrades, as the language capabilities of

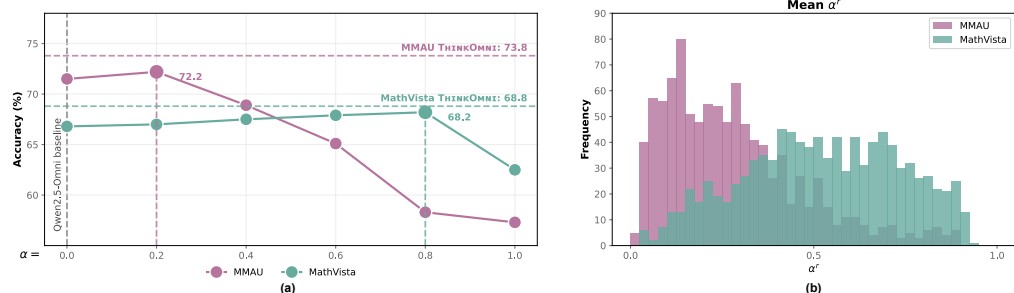

Figure 7: **Ablation on guidance weight.** Left: Performance varies with the constant guidance weight $\alpha$. Each task's optimal $\alpha$ range differs, with $\alpha = 0$ as the OLLM baseline. Right: THINKOMNI uses adaptive dynamic weights, and the dynamic $\alpha^r$ shows a similar distribution shift, indicating that stepwise contrastive scaling can flexibly adapt to different task requirements.

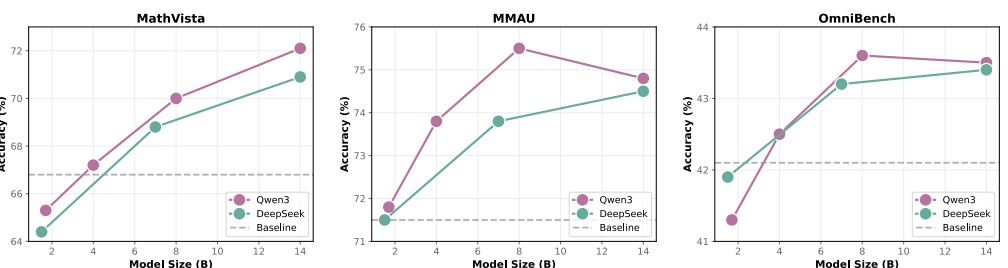

Figure 8: **LRM-as-a-Guide performance scaling.** We replace the Guide LRM on several benchmarks to study the impact of different LRM sizes and different LRM series on the performance of our method. The baseline refers to the performance of the Qwen2.5-Omni-7B

LRM and the base OLLM are insufficiently matched. A sufficiently strong LRM positively impacts all benchmarks, with larger LRMs generally yielding better results than smaller ones. However, increasing the size to Qwen3-14B does not lead to further improvements on MMAU and OmniBench, suggesting that enhanced reasoning ability has a limited effect on these tasks, although the results still surpass the baseline.

## 4.5 ANALYSIS

**Qualitative analysis** Fig. 9 illustrates the generation process of THINKOMNI, where darker colors indicate greater LRM contributions to each token. These tokens, often logical connectives (e.g., "but", "Therefore") and key terms (e.g., "traditional", "common"), are evenly distributed, showing that LRM consistently guides reasoning rather than merely supplementing OLLM. This highlights LRM's role in analyzing multi-modal clues and driving logical inference. In contrast, lighter tokens are mainly function words and specific terms reflecting multi-modal content, suggesting that OLLM focuses on retrieving information and constructing fluent responses under LRM's guidance. For more cases, see Appendix D.

**Failure case analysis** We identified several representative failure cases from the response. 1) THINKOMNI demonstrates correct multimodal perception, but conflicting information within the input leads to erroneous reasoning. As shown in Fig. 10(a), the model correctly recognizes that the highest visible marking on the beaker is 400ml. However, due to the beaker being labeled as 600ml, the model incorrectly infers that only a portion of the beaker is visible in the image, resulting in the wrong final answer. 2) Insufficient perceptual ability and limited sensitivity to subtle differences in the input lead to incorrect answers. As illustrated in Fig. 10(b), the model fails to accurately detect the actual onset of the drum kit in the audio and mistakenly identifies the beginning of the audio as the start of the drum kit's performance, ultimately producing the wrong answer.

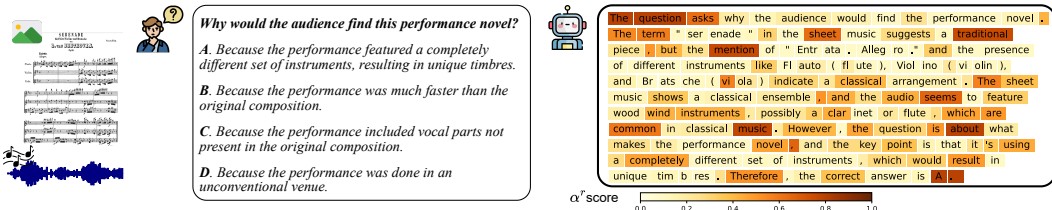

Figure 9: **An OmniBench case study.** This case study visualizes the reasoning process of ThinkOmni-Qwen2.5-Omni-7B, highlighting the stepwise contrastive scaling coefficient $\alpha^r$.

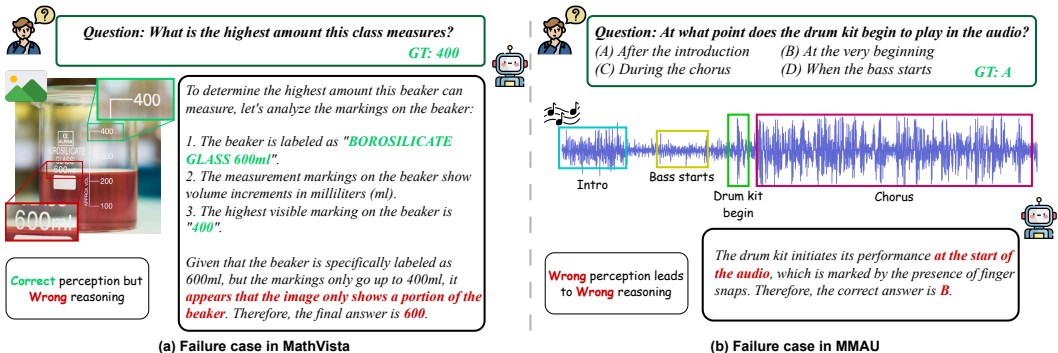

Figure 10: **Failure case study.** (a) ThinkOmni makes a reasoning error due to conflicting information between the visible beaker markings (400ml) and the label (600ml). (b) ThinkOmni fails because it cannot accurately detect the drum kit's true start time in the audio.[1]

**Efficiency analysis** Although our method introduces some additional computation, it remains efficient. We measured generation latency on an H800-80G GPU with KV cache in the generate stage, benchmarking VCD (Leng et al., 2024), ProxyTuning (Liu et al., 2024), and ThinkOmni using 100 random OmniBench (Li et al., 2024) samples. VCD performs two forward passes per step, while ProxyTuning and ThinkOmni require three. As shown in Tab. 3, our method (7B+7B setting) incurs $1.38\times$ in the prefill stage (first token generation) and $2.88\times$ in the generate stage (response generation utilizing KV cache). Importantly, during decoding, the guiding model in ThinkOmni processes only text, which helps to reduce latency.

Table 3: **Generation latency comparison.** Prefill Time: first token generation latency; Generate Time: response generation with KV cache. Results are averaged over 100 samples from OmniBench.

| Guidance Decoding Method | Model Size | Prefill Time ↓ | Generate Time ↓ |
|---|---|---|---|
| None (Baseline) | 7B | 0.138s | 0.025s |
| VCD (Leng et al., 2024) | 7B | 0.262s (1.89×) | 0.050s (2.00×) |
| ProxyTuning (Liu et al., 2024) | 7B + 3B + 3B | 0.406s (2.94×) | 0.086s (3.44×) |
| ThinkOmni (Ours) | 7B +7B | 0.191s (1.38×) | 0.072s (2.88×) |
| ThinkOmni (Ours) | 7B +1.5B | 0.191s (1.38×) | 0.069s (2.76×) |

## 5 RELATED WORK

**Omni-modal Large Language Models** With the rapid advancement of large language models, expanding their capabilities to omni-modal domains has become a key focus. Omni-modal Large Language Models (OLLM) align and process information from multiple modalities, capturing richer semantics and context than single-modal systems. Proprietary models (Hurst et al., 2024; Deepmind, 2025) demonstrate impressive real-time multi-modal interaction, while open-source efforts such as

---

[1]For the reader's understanding, the colored markings in Fig. 10 are visual aids added afterward and were not provided to the model.

Qwen2.5-Omni (Xu et al., 2025; Li et al., 2025b; Liu et al., 2025c; Yang et al., 2025b; Fu et al., 2025; Xie & Wu, 2024; Chen et al., 2025) are quickly closing the gap in modality alignment and deployment.

**Large Reasoning Model**   Recent advances in large-scale reasoning models, such as o1 (OpenAI, 2025) and DeepSeek-R1 (Guo et al., 2025), highlight the challenge of robust general reasoning. Early methods used supervised fine-tuning with chain-of-thought data (Xu et al., 2024; Linger et al., 2025), while recent work leverages reinforcement learning (Shao et al., 2024; Xiaomi et al., 2025; Team et al., 2025; Yu et al., 2025; Tan et al., 2025; Zhu et al., 2026) for autonomous reasoning. As OLLMs tackle complex cross-modal reasoning, bridging perception and reasoning remains a core challenge.

**Decoding-time Algorithm**   Decoding-time algorithms refine language model outputs at inference in a training-free manner. Contrastive Decoding (O'Brien & Lewis, 2023) improves long-form generation by avoiding degenerate outputs, and Visual Contrastive Decoding (Leng et al., 2024) reduces hallucination via visual input perturbation. ProxyTuning (Liu et al., 2024) combines expert outputs and injects knowledge from finetuned models. ProxyThinker (Xiao et al., 2026) extends ProxyTuning to multi-modal reasoning tasks. Beyond visual modalities, THINKOMNI adaptively integrates reasoning with omni-modal perception, achieving a flexible fusion of fast and slow thinking.

## 6 CONCLUSION

We present THINKOMNI, a training-free inference-time framework that achieves robust and generalizable reasoning enhancement by introducing an off-the-shelf LRM guide decoding with a stepwise adaptive scaling mechanism. Across six challenging multi-modal benchmarks, it delivers consistent gains, often matching or surpassing reinforcement-fine-tuned models, suggesting a general and extensible paradigm for omni-modal reasoning.

**Limitation**   THINKOMNI requires shared vocabularies between the OLLM and LRM for logit fusion and introduces extra inference overhead due to additional forward passes. Nevertheless, we believe our approach offers valuable insights for bridging the gap between multi-modal and textual LLMs and provides a sustainable direction for future LLM improvements.

## AKNOWLEDEGMENT

This work was done during the research internship of Yiran Guan, Sifan Tu, Linghao Zhu, and Dingkang Liang at Xiaomi Inc. MiLM Plus team.

This work was supported by the National Natural Science Foundation of China (NO. 62441615, 62225603, 62576147).

## ETHICS STATEMENT

This work introduces a training-free framework using only publicly available models (Qwen2.5-Omni (Xu et al., 2025), Omni-R1 (Zhong et al., 2025), DeepSeek-R1 (Guo et al., 2025), Qwen3 (Yang et al., 2025a)) and benchmarks (see Appendix B), without collecting new human, biometric, or sensitive data. Risks stem from inherited biases, possible hallucinated cross-modal attributions, and over-trust in generated reasoning chains; the method does not ensure factuality or safety in high-stakes domains. Before deployment, we advise bias auditing, human oversight, and external safety / factuality filters. Environmental impact is reduced relative to finetuning because no additional training is performed.

## REPRODUCIBILITY STATEMENT

Reproducibility is supported by: 1) the explicit inference formulation (Eq. 3 and Eq. 5); 2) unified decoding hyperparameters (see Appendix B); 3) public benchmark splits and the two-stage answer extraction pipeline follow Xiao et al. (2026); 4) hardware specification (80G VRAM GPU, KV cache enabled). The code will be coming up soon upon acceptance.

## LLM USAGE

LLMs were used only for 1) minor code scaffolding / refactoring (boilerplate, log parsing), 2) linguistic polishing after technical content was finalized, and 3) answer string normalization in the evaluation pipeline (format extraction, not scoring). They were not used for ideas, model / method design, experiments, analyses, claims, or interpretations. The researchers authored, validated, and cross-checked all algorithms, equations, results, and conclusions. Every LLM-assisted output was manually reviewed to prevent hallucination. Thus, LLM involvement provides no creative contribution and does not affect the authenticity or reliability of the paper.

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

## A  A DETAILED EXPLANATION FOR THINKOMNI

At each decoding step $t$, the Omni-modal Large Language Model (OLLM) generates two token probability distributions: conditioned on the full omni-modal input and the textual only input. In contrast, the Large Reasoning Model (LRM) outputs a single distribution based solely on the textual input.

We would like to decide how much of the next-token decision should be driven by perception $(P_O^{(t)})$ and how much by abstract reasoning $(P_R^{(t)})$. To gauge the *disagreement* between these signals, we compute two Jensen–Shannon divergences

$$D_R^{(t)} \;=\; \mathrm{JS}\big(P_R^{(t)} \parallel P^{(t)}\big), \qquad D_P^{(t)} \;=\; \mathrm{JS}\big(P_O^{(t)} \parallel P^{(t)}\big).$$

$D_R^{(t)}$ is large when the *reasoner* (LRM) wants a different token than the perceiver (OLLM text-only); $D_P^{(t)}$ is large when masking the multi-modal content changes the OLLM's belief is crucial, i.e. when *perception*.

1. If the current step mainly requires **perception** (e.g. reading a label in the image or detecting a sound), masking the modalities hurts the OLLM, so $D_P^{(t)}$ is large while $D_R^{(t)}$ stays small. Consequently $\alpha_t^r \approx 0$ and the model trusts the *perception* logits.

2. If the step calls for **reasoning** (e.g. algebraic manipulation after the visual information is extracted), the text-only LRM disagrees with $P^{(t)}$, so $D_R^{(t)}$ dominates and $\alpha_t^r \approx 1$. The generation is therefore guided by the stronger logical signal from the LRM.

3. For mixed cases, $\alpha_t$ smoothly interpolates between the two extremes, allowing the decoder to blend perception and reasoning in real time.

## B  EXPERIMENT DETAILS

**Hyperparameter Settings**   During our experiments, we used the following inference parameters to ensure standard model outputs and to prevent performance degradation and endless repetitions, following (Yang et al., 2025a): a `temperature` of 0.6, a `top_p` value of 0.95, a `repetition_penalty` of 1.03, and a `max_new_tokens` of 4096. In addition, all LRMs append a `<think>` tag to the end of the prompt during inference to ensure the reasoning state is activated (Guo et al., 2025).

**Dynamic Weight and Mixing.**   Given the calculated divergences $D_R$ and $D_P$, we determine the reasoning weight $\alpha_t^r$ by measuring the reasoning surplus, clamped to $[0, 1]$:

$$\alpha_t^r = \mathrm{clip}\left(D_R - D_P, 0, 1.0\right). \tag{6}$$

To ensure stability during the initial phase, we apply a linear warmup for the first $T_{\mathrm{warm}} = 5$ steps:

$$\alpha_t^r \leftarrow \min\left(\alpha_t^r, 0.1 \cdot t\right). \tag{7}$$

Finally, the mixed logits are computed efficiently via the following implementation, which integrates the base, reasoning, and negative distributions:

$$\hat{z}_t = (2 - \alpha_t^r) \cdot M_O(x_{<t}, O) + \alpha_t^r \cdot M_R(x_{<t}) - M_O(x_{<t}). \tag{8}$$

**Benchmarks Details**   The datasets used in our experiments include MathVista (Lu et al., 2024), MathVision (Wang et al., 2024), MathVerse (Zhang et al., 2024), MMAU (Sakshi et al., 2025), Daily-Omni (Zhou et al., 2025) and OmniBench (Li et al., 2024).

- **MathVista** (Lu et al., 2024): We evaluate on the test-mini split of MathVista (1,000 samples), a unified benchmark for mathematical reasoning in visual contexts, which includes three newly introduced datasets (IQTest, FunctionQA, and PaperQA), as well as 9 MathQA and 19 VQA datasets from previous work.

- **MathVision** (Wang et al., 2024): We evaluate on the MathVision dataset (3,040 samples), a curated collection of high-quality mathematical problems with visual contexts. Covering 16 mathematical disciplines and five difficulty levels, MATHVision offers a comprehensive and diverse benchmark for assessing the mathematical reasoning abilities of LMMs.

- **MathVerse** (Zhang et al., 2024): We evaluate on the test-mini split of MathVerse (3,940 samples), a visual math benchmark with 2,612 multi-subject problems and diagrams. Each issue is annotated into six multi-modal versions, totaling 15K samples, to assess MLLMs' understanding of visual information in math reasoning.

- **MMAU** (Sakshi et al., 2025): We evaluate on the test-mini split of MMAU (1,000 samples), which consists of 10K audio clips with natural language questions and answers across speech, sounds, and music. Covering 12 retrieval and 15 reasoning types, MMAU challenges models with expert-level, domain-specific audio understanding and reasoning.

- **Daily-Omni** (Zhou et al., 2025): We evaluate on Daily-Omni (1,197 samples), which contains 684 videos from 11 YouTube categories and questions requiring integration of audio, visual, and textual information. The benchmark covers 30-second and 60-second videos to assess multi-modal reasoning abilities.

- **OmniBench** (Li et al., 2024): We evaluate on OmniBench (1,142 samples), which covers a wide range of reasoning and cognitive skills, from perception to complex reasoning. Tasks include object recognition, temporal and spatial reasoning, symbolic and quantitative processing, and various audio types, including speech, sound events, and music.

## C    FUTURE WORK

THINKOMNI represents an attempt to introduce omni-modal capabilities based on textual reasoning abilities. We plan to explore additional modalities, such as 3D point clouds, protein structures, and reasoning applications in image/video generation scenarios. Moreover, we are also curious about "what truly works during the reasoning process", which is of great significance for understanding why reasoning abilities in the textual domain can generalize to a wider range of modalities.

## D MORE CASES

*Please answer the following question based on the given image and audio:*
*where this person in white close might be.*
*Please choose only one answer from the following options:*
*A. woman in jail*
*B. woman in shelter*
*C. man in court*
*D. man in police station*

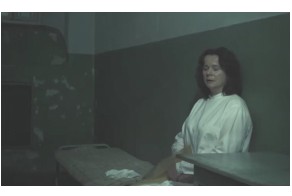 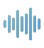

Audio content:
(Key to open the door + conversation + close the door)
I'll come back when the paperwork is complete.

*Answer:*
*The dim lighting, bare walls, and the presence of a simple bed suggest a confined space typical of a jail cell. The sound effects of metal clanging and a door slamming reinforce the idea of a prison environment. The person's attire is modest, which could indicate a prisoner.*

*While shelters can have similar settings, the specific sound effects and the presence of a bed rather than a cot or sleeping bag make it less likely. Shelters typically have more varied furnishings and may not have the same level of confinement.*

*The setting does not resemble a courtroom, which usually has a bench, desks, and more formal attire. The sounds heard do not match those of a court proceeding.*

*A police station might have similar elements, but the presence of sound effects and the specific description of the room as a cell point more towards a jail rather than a police station. Police stations often have more varied spaces and different types of furniture.*

*The answer is: A.*

Figure 11: Omni-modal Reasoning.

*Question: What is the most likely environment and time of day for these sounds?*
*Choices:*
*(A) Urban street at night*
*(B) Quiet home interior in the morning*
*(C) Busy office during lunch hour*
*(D) Forest trail at dusk*

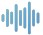

Audio content:
The audio includes the sounds of cats and birds, and the ambient sound is relatively open.

*Answer: The cat's meowing, combined with bird chirping, suggests a quiet setting where both animals are active, which is more common in a natural environment like a forest trail during dusk. The presence of bird chirping also indicates daylight, eliminating the possibility of it being at night or during a busy office hour. The quietness of the environment points towards a home interior, but the presence of birds suggests an outdoor setting, making the forest trail at dusk a less likely choice. Therefore, the most probable environment is a quiet home interior in the morning.*

*The answer is: B.*

Figure 12: Audio Reasoning.

*Question: The derivative of f(x) at x=2 is _____ that at x=5*
*Choices:*
*(A) larger than*
*(B) equal to*
*(C) smaller than*

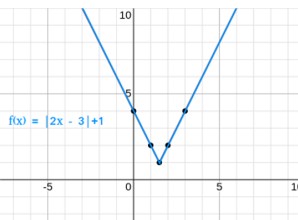

*Answer: To determine the relationship between the derivative of f(x) at x=2 and x=5, we need to analyze the graph of the function f(x) = |2x - 3| + 1.*

*1. **Identify the critical point**: The function f(x) = |2x - 3| + 1 has a V-shape, which means it has a vertex where the expression inside the absolute value changes sign. This occurs at 2x - 3 = 0, which gives x = 3/2.*

*2. **Analyze the slope on either side of the vertex**:*
  *- For x < 3/2, the expression inside the absolute value is negative, so f(x) = -(2x - 3) + 1 = -2x + 3 + 1 = -2x + 4. The slope here is -2.*
  *- For x > 3/2, the expression inside the absolute value is positive, so f(x) = (2x - 3) + 1 = 2x - 3 + 1 = 2x - 2. The slope here is 2.*

*3. **Evaluate the slopes at x=2 and x=5**:*
  *- At x=2, which is greater than 3/2, the slope is 2.*
  *- At x=5, which is also greater than 3/2, the slope is still 2.*

*Since the slope (derivative) is the same for both x=2 and x=5, the derivative of f(x) at x=2 is equal to that at x=5.*

*Therefore, the correct answer is B.*

Figure 13: Visual Reasoning.

