# OpenReview forum: "ThinkOmni: Lifting Textual Reasoning to Omni-modal Scenarios via Guidance Decoding"
_ICLR.cc/2026/Conference — ICLR 2026 Poster_

### Official Review · Reviewer_2VXa · 2025-10-26

**Soundness:** 3
**Presentation:** 3
**Contribution:** 3
**Rating:** 6
**Confidence:** 4

**Summary:**

The paper introduces ThinkOmni, a training-free framework designed to enhance the reasoning capabilities of Omni-modal Large Language Models by leveraging the strengths of text-based Large Reasoning Models (LRMs). The core problem is that while OLLMs excel at perceiving diverse inputs , they lack the advanced reasoning skills of specialized LRMs. Bridging this gap usually requires costly fine-tuning or reinforcement learning, demanding large datasets and significant computation. It doesn't need training by using an LRM to guide the OLLM's decoding process at inference time. Experiments show ThinkOmni significantly boosts the performance of OLLMs on challenging multimodal reasoning benchmarks.

**Strengths:**

1. It avoids the need for expensive data collection, annotation, and computationally intensive fine-tuning or RL, because of its training-free nature.

2. it effectively combines the perceptual strengths of OLLMs with the reasoning strengths of state-of-the-art LRMs in a modular way.

3. it demonstrates substantial performance gains on difficult benchmarks, showing the practical effectiveness of the approach.

**Weaknesses:**

1. running both an OLLM and an LRM during inference will inevitably increase computational cost and latency compared to running a single model, even if optimized.

2. the framework's performance ceiling is inherently tied to the capabilities of the chosen LRM. Errors or limitations in the LRM's reasoning will directly impact the guided OLLM.

3. the LRM guide operates on a textual representation of the omni-modal input. Complex non-textual information (e.g., intricate spatial relationships in an image, temporal dynamics in a video, nuances in audio) might be lost or distorted in this conversion, limiting the LRM's ability to provide accurate guidance.

**Questions:**

1. It's unclear how the framework robustly handles situations where the OLLM's direct perception strongly contradicts the LRM's reasoning based on the potentially lossy text representation.

2. How significant is the increase in inference time and computational cost when using ThinkOmni compared to running the base OLLM alone?

3. How does the framework handle tasks where the crucial reasoning step relies heavily on non-translatable visual, auditory, or temporal patterns (e.g., recognizing subtle visual defects, interpreting musical harmony, understanding complex physical interactions in a video)? Can specific examples of failure cases due to information loss in text conversion be provided?

4. What happens when the guiding LRM makes a fundamental reasoning error? Does the Stepwise Contrastive Scaling mechanism effectively detect and mitigate the impact of incorrect guidance, or can it lead the OLLM further astray?

5. How sensitive is the performance to the quality of the text summarization/description of the omni-modal input fed to the LRM? Who generates this text, and how is its fidelity ensured?

6. It mentions "maintaining decoding efficiency" (line 104). How is this efficiency achieved when two large models are involved in the decoding loop?

7. How does ThinkOmni perform on tasks requiring common-sense reasoning grounded in multimodal contexts, rather than specialized mathematical or logical reasoning?

---

> ### Author Response · Authors · 2025-11-20
> **Response (1/2)**
>
> Thank you for your positive recognition of our training-free framework and its practical effectiveness in enhancing multimodal reasoning; we hope our response can thoroughly address your concerns.
>
> > **W1 & Q2 & Q6:** Running both an OLLM and an LRM ... How significant is the increase ... compared to running the base OLLM alone? ... How is this efficiency achieved when two large models are involved in the decoding loop?
>
> Compared to using a single OLLM, our method does incur higher resource consumption as it involves both an OLLM and an LRM in the decoding loop. However, this trade-off results in significant performance gains that would otherwise require much larger models, resulting in greater inference and training overhead, or access to extensive training data.
>
> For a fair comparison, we report the inference speed in `Tab. 3`  based on the serial model invocation setup used in the paper. In practice, the three forward passes in ThinkOmni can be executed in parallel. Here, we additionally report the results when parallelizing these three forward passes:
>
> | Setting              | Generate Time (s)       |
> | -------------------- | -------------------- |
> | Qwen2.5-Omni-7B      | 0.025                |
> | ThinkOmni (serial)   | 0.072 ($\times2.88$) |
> | ThinkOmni (parallel) | 0.039 ($\times1.56$) |
>
> As noted in `line 91` of the paper, our decoding efficiency is competitive with **other guidance-based algorithms**. From `Tab. 3`, ThinkOmni achieves lower prefill time (since multimodal information is processed only once), and our generate time is superior to ProxyTuning, which requires three separate forward passes.
>
> Moreover, there remains room for engineering optimizations to further improve speed, such as speculative decoding [1, 2] based on the original OLLM. We believe these strategies will help mitigate the additional computational cost while maintaining strong performance.
>
> [1]Leviathan et al. "Fast Inference from Transformers via Speculative Decoding." ICML 2023.
>
> [2] Yuan et al. "Speculative Contrastive Decoding." ACL 2024.
>
> > **Q1:** It's unclear how the framework robustly handles situations where the OLLM's direct perception strongly contradicts the LRM's reasoning based on the potentially lossy text representation.
>
> A simple example illustrates our mechanism. In every multi-modal task, the direct perception of multi-modal information is often an action that conflicts with the LRM’s output.
>
> Let's revisit the process in which logit are used to generate the sampling distribution `Eq. (6)`:
> $$
> \hat{P} = \mathrm{Softmax}\Big[\,
>     z_{base}
>     + \alpha^r_t \cdot \big( z_{+} - z_{-} \big)
>     + \alpha^p_t \cdot \big( z_{base} - z_{-} \big)\Big],
> $$
> Here, $z_{+} $ is the logit from the LRM, and $ z_{-} $ is the logit from the OLLM without the multimodal information condition.
>
> During perception actions, both $ z_{+} $ and $ z_{-} $ lack direct multimodal information, so their interference regarding perception is reduced by subtracting their logits. Additionally, thanks to the Stepwise Contrastive Scaling mechanism, the reasoning factor $ \alpha^r_t $ (which is relatively small during perception) further attenuates the impact of conflicting information from LRM.
>
> Once successful decoding occurs and multimodal information is obtained, the LRM can then use the updated context to continue reasoning and decoding subsequent tokens.
>
> > **W2 & Q4:** The framework's ... Errors or limitations in the LRM's reasoning will directly impact the guided OLLM. What happens when the guiding LRM makes a fundamental reasoning error? Does the Stepwise Contrastive Scaling mechanism ... ?
>
> We agree with you that the framework’s overall performance is indeed affected by the quality of the LRM, since our approach relies on leveraging off-the-shelf LRM models for reasoning. This dependency is inherent to our design, which aims to combine existing reasoning models with perceptual models in a flexible manner.
>
> The Stepwise Contrastive Scaling mechanism is not designed to correct errors within the LRM itself. Its primary purpose is to balance perception and reasoning by mixing logits at each step, effectively decoupling the perception process from LRM’s reasoning influence. This helps prevent reasoning errors from affecting multimodal understanding during perception steps. However, when reasoning is required, our framework still depends on the LRM, so fundamental mistakes from the LRM can impact the final output. Our ablation studies (as shown in `Fig. 8`) further show that the quality of the LRM remains crucial.

---

> ### Author Response · Authors · 2025-11-20
> **Response (2/2)**
>
> > **W3 & Q3:** The LRM guide operates ... Complex non-textual information ... limiting the LRM's ability to provide accurate guidance. How does the framework handle ... non-translatable visual, auditory, or temporal patterns ... examples of failure cases due to information loss in text conversion be provided?
>
> Thank you for your question. We have carefully analyzed the unique failure modes of our method and included relevant discussion in the latest version of our paper. As shown in Fig. 10(a) (or access directly via [**this_link**](https://anonymous.4open.science/r/ThinkOmni-8C48/Failure%20Case%20Study.png)), the OLLM accurately perceives the content, but the LRM introduces incorrect reasoning. For example, the model identifies the beaker’s capacity as 600ml and the highest visible marking as 400ml, which leads to the mistaken conclusion that only part of the beaker is shown. This type of limitation is common when combining multimodal models and language models for reasoning, as noted in [1, 2, 3].
>
> Unlike approaches that rely solely on language-based descriptions for LRM reasoning (essentially using only the top-1 logits), our method compares tokens at the logits level, providing richer information. This is particularly beneficial when recognizing ambiguous objects, since logits reflect not only the most likely category but also less likely alternatives. As a result, logits mixing enables more nuanced multimodal understanding. Our method outperforms the Caption-then-Answer baseline across multiple benchmarks:
>
> | Setting             | MathVista | MMAU | OmniBench |
> | ------------------- | --------- | ---- | --------- |
> | Caption-then-Answer | 61.0      | 59.7 | 32.3      |
> | ThinkOmni           | 68.8      | 73.8 | 43.2      |
>
> [1] Guo et al. "Decoupled Visual Interpretation and Linguistic Reasoning for Math Problem Solving" ICCV 2025.
>
> [2] Ossowski et al. "COMMA: A Communicative Multimodal Multi-Agent Benchmark." TMLR 2025.
>
> [3] Guo et al. "Reasoning-Aligned Perception Decoupling for Scalable Multi-modal Reasoning." arXiv preprint 2506.04559.
>
> > **Q5:** How sensitive is the performance to the quality of the text summarization/description of the omni-modal input fed to the LRM? Who generates this text, and how is its fidelity ensured?
>
> The OLLM generates the description and perception of the input content. The perceptual ability of the OLLM directly impacts the quality of the final answer. To illustrate this, we show the impact of OLLM’s perceptual capability on the overall results:
>
> | LRM      | OLLM            | MathVista | MMAU | OmniBench |
> | -------- | --------------- | --------- | ---- | --------- |
> | Qwen3-8B | Qwen2.5-Omni-3B | 58.1      | 70.6 | 39.5      |
> | Qwen3-8B | Qwen2.5-Omni-7B | 70.2      | 75.5 | 43.6      |
> | Qwen3-8B | Omni-R1-7B      | 71.3      | 75.4 | 43.4      |
>
> As shown above, models with stronger perceptual capability (7B) achieve significantly better results than the 3B model. Meanwhile, Omni-R1-7B, which is trained with reinforcement learning to enhance reasoning ability, only shows a noticeable improvement on MathVista. This dataset requires more reasoning, compared to the base Qwen2.5-Omni-7B model. This suggests that reasoning ability primarily influences tasks that require complex inference, whereas perceptual ability is essential for all tasks.
>
> In addition, our newly added failure cases (see `Fig. 10(b)` or access directly via **[**this_link**](https://anonymous.4open.science/r/ThinkOmni-8C48/Failure%20Case%20Study.png)**) further demonstrate that limited perceptual capability can directly lead to incorrect answers. Our method focuses on cases where perception is correct but reasoning is lacking. To improve the quality of multimodal explanations, we primarily rely on using stronger OLLMs.
>
> > **Q7:** How does ThinkOmni perform on tasks requiring common-sense reasoning grounded in multimodal contexts, rather than specialized mathematical or logical reasoning?
>
> We have supplemented our experiments with results on MMMU-Pro, MMMU, and MME, which are benchmarks specifically designed to evaluate such capabilities. As shown in the table below, ThinkOmni achieves consistent improvements over the Qwen2.5-Omni-7B baseline when guided by Qwen3-8B:
>
> | Setting         | MMMU-Pro-overall | MMMU-val | MME_sum |
> | --------------- | ---------------- | -------- | ------- |
> | Qwen2.5-Omni-7B | 36.47            | 50.67    | 2304    |
> | +Qwen3-8B       | 39.02            | 53.78    | 2359    |
>
> These results demonstrate that ThinkOmni not only excels in specialized mathematical and logical reasoning tasks, but also delivers substantial gains on benchmarks requiring common-sense reasoning in multimodal settings.

---

> > ### Comment · Reviewer_2VXa · 2025-11-26
> >
> > Thanks the authors for their reply. I will increase the soundness to 4 and confidence to 5.

---

> > > ### Author Response · Authors · 2025-11-26
> > >
> > > Thank you for taking the time to provide such thoughtful feedback on our paper. We are pleased that our responses and revisions have been well received, and we appreciate that you have increased the soundness and confidence scores of our work.

---

### Official Review · Reviewer_kNcB · 2025-11-01

**Soundness:** 4
**Presentation:** 4
**Contribution:** 4
**Rating:** 6
**Confidence:** 4

**Summary:**

This paper proposes the ThinkOmni framework, which aims to enhance the reasoning capabilities of multimodal large language models (OLLMs) by utilizing off-the-shelf large reasoning models (LRMs). By introducing two innovations—LRM-as-a-Guide and Stepwise Contrastive Scaling—ThinkOmni can flexibly balance perceptual and inference signals without additional training, thereby improving multimodal reasoning performance. Experimental results show that ThinkOmni achieves significant performance improvements on multiple benchmark tasks, outperforming existing models that rely on reinforcement fine-tuning.

**Strengths:**

1. Innovative Framework: The paper proposes ThinkOmni, a training-free, decoding-time method that cleverly integrates a Large Reasoning Model (LRM) into an omni-modal LLM (OLLM) pipeline. The LRM-as-a-Guideand Stepwise Contrastive Scaling mechanisms are novel and well-motivated.
2. Strong Empirical Results: The method shows consistent and significant improvements across six diverse omni-modal reasoning benchmarks. The gains are particularly notable given that no additional training is required.

**Weaknesses:**

1. Limited Focus on Math Domain:  The benchmark is currently restricted to the math domain at the image-level. It would be beneficial to expand the experiments to include perception-level tasks, such as MME, as well as reasoning benchmarks like MMLU, to provide a more comprehensive evaluation.

2. Confusion Regarding Figure 4: There is some confusion regarding the notation in Figure 4, specifically the terms \( x_{<t} \) and \( x_{<t+1} \). These appear to suggest a multi-turn conversational setup, which warrants further clarification.

3. Minor Performance Gain Relative to ProxyTuning:  The observed performance gain is relatively minor when compared to ProxyTuning, which calls for a deeper analysis of the effectiveness of the proposed approach in this context.

**Questions:**

Please see weakness.

---

> ### Author Response · Authors · 2025-11-20
>
> Thank you for acknowledging the innovative framework and robust empirical results presented in our work. We hope that our response effectively addresses your concerns.
>
> > **W1:** Limited Focus on Math Domain ... It would be beneficial to expand the experiments to include perception-level tasks, such as MME, as well as reasoning benchmarks like MMLU ...
>
> In accordance with your suggestion, we have supplemented our results with experiments on MME and MMMU-pro, MMMU (we assume you intended to refer to MMMU, as MMLU is an evaluation metric for language models). The corresponding results are presented below.
>
> | Setting         | MMMU-Pro-overall | MMMU-val | MME-sum |
> | --------------- | ---------------- | -------- | ------- |
> | Qwen2.5-Omni-7B | 36.47            | 50.67    | 2304    |
> | +Qwen3-8B       | 39.02            | 53.78    | 2359    |
>
> The experimental settings are consistent with `Section 4.2` of our paper. As the experimental results show, our method delivers robust improvements in common-sense reasoning, whereas the performance gains in MME are relatively modest.
>
> > **W2:** Confusion Regarding Figure 4 ... $x_{<t}$ and $x_{<t+1}$ . These appear to suggest a multi-turn conversational setup, which warrants further clarification.
>
> The figure illustrates the next token prediction process, rather than a multi-turn conversational setup. We have revised `Fig. 4` in the paper to provide clearer explanations for ambiguous indications. The notations  $x_{<t} = (x_1, x_2, \ldots, x_{t-1})$ denotes the sequence of tokens used for the next token prediction. Specifically, $x_{<t+1}$ is formed by appending the newly generated token $x_t$ to the conditioning sequence $x_{<t}$, following the standard generation process of LLMs rather than a multi-turn dialogue. `Fig. 4` shows how ThinkOmni incorporates the influence of full-modal input and reasoning capability at each token generation step to determine the mixing ratio of logits for the output.
>
> > **W3:** Minor Performance Gain Relative to ProxyTuning: The observed performance gain is relatively minor when compared to ProxyTuning...
>
> Our method outperforms ProxyTuning by more than one percentage point on both the MathVista and MMAU benchmarks. This improvement exceeds the performance gain that ProxyTuning achieves over the baseline.
>
> The performance of ProxyTuning depends on an OLLM that has been specifically fine-tuned for the target domain or modality. However, this reliance on specialised, task-specific models creates a significant bottleneck, as costly retraining and data collection are required for each new domain. This limits their general applicability. In contrast, our approach is fundamentally more flexible and scalable. It uses a powerful LRM that does not require any domain-specific fine-tuning. This design makes our framework inherently more extensible. It can simply be extended by integrating stronger off-the-shelf LRMs as they become available.

---

### Official Review · Reviewer_myT4 · 2025-11-02

**Soundness:** 3
**Presentation:** 3
**Contribution:** 3
**Rating:** 6
**Confidence:** 5

**Summary:**

The paper proposes ThinkOmni, a training-free framework that enhances omni-modal reasoning by leveraging off-the-shelf Large Reasoning Models (LRMs) to guide the decoding process of Omni-modal Large Language Models (OLLMs). The method consists of two main components: LRM-as-a-Guide, which uses textual reasoning models to influence OLLM outputs via logit mixing, and Stepwise Contrastive Scaling, which dynamically balances perception and reasoning signals during inference. The approach is evaluated on six omni-modal benchmarks, showing consistent improvements over baseline OLLMs and competitive performance against reinforcement fine-tuned models.

**Strengths:**

- The proposed Stepwise Contrastive Scaling mechanism is a meaningful contribution, enabling adaptive tuning of guidance weights without manual hyperparameter search.
- Extensive experiments across six benchmarks demonstrate the generality and scalability of the method.
- The framework is model-agnostic and can be applied to various OLLM and LRM combinations, enhancing its potential impact.
- The idea of leveraging pre-trained LRMs to enhance OLLM reasoning without additional training is novel and practical, especially given the high cost of omni-modal fine-tuning.

**Weaknesses:**

- The approach requires external LLMs to guide the response. This led to doubts about where the performance gain comes from. Does LLM introduce extra information for solving the question?
- Although training-free, the approach incurs non-trivial inference overhead due to multiple forward passes per decoding step, which may hinder real-time deployment.

**Questions:**

- What is the theoretical justification for using Jensen-Shannon divergence to measure disagreement between distributions?  Were other metrics explored?
- How does the method perform when the OLLM and LRM are from different model families or have significant architectural differences?
- How does ThinkOmni handle cases where the LRM provides incorrect reasoning?  Is there a risk of propagating textual reasoning errors into omni-modal outputs?

---

> ### Author Response · Authors · 2025-11-20
> **Response (1/2)**
>
> Thank you for your recognition of our approach and motivation. We will do our best to address your concerns.
>
> > **W1:** The approach requires external LLMs to guide the response. This led to doubts about where the performance gain comes from. Does LLM introduce extra information for solving the question?
>
> We would like to clarify that our approach does not rely on external LLMs to inject additional knowledge beyond what is already available. Instead, the primary role of the LRM is to guide the reasoning process, rather than to provide supplementary information.
>
> We conducted controlled experiments comparing three settings: (1) OLLM baseline, (2) OLLM combined with Qwen3 8B without the "think" mode, and (3) OLLM combined with Qwen3 with the "think" mode. These experiments were designed to isolate the effect of the reasoning process, under the assumption that the same LRM has access to the same knowledge in both modes. The results are shown below:
>
> | Setting           | MathVista | MMAU | OmniBench |
> | ----------------- | --------- | ---- | --------- |
> | Qwen2.5-Omni-7B   | 66.8      | 71.5 | 42.1      |
> | +Qwen3 (no think) | 67.1      | 71.3 | 42.1      |
> | +Qwen3 (think)    | 70.2      | 75.5 | 43.6      |
>
> As shown, the significant performance improvement is observed only when the "think" mode is enabled, confirming that the gains are primarily attributable to the guided reasoning process rather than the introduction of extra knowledge.
>
> > **W2:** Although training-free, the approach incurs non-trivial inference overhead due to multiple forward passes per decoding step, which may hinder real-time deployment.
>
> As you have noted, ThinkOmni incurs additional inference overhead compared to the original OLLM. This increased computational cost is primarily due to the need for multiple forward passes per decoding step. We acknowledge this trade-off, as our approach leverages both the OLLM and LRM throughout the decoding process to achieve stronger multimodal reasoning capabilities.
>
> For a fair comparison, the inference speed reported in our paper is based on a serial model invocation setup. However, in practical deployment, the computation of logits for each model can be parallelized during inference, which substantially reduces the time overhead in real-world scenarios. We conducted a simple parallel deployment, and the results are as follows.
>
> | Setting              | Generate Time  (s)      |
> | -------------------- | -------------------- |
> | Qwen2.5-Omni-7B      | 0.025                |
> | ThinkOmni (serial)   | 0.072 ($\times2.88$) |
> | ThinkOmni (parallel) | 0.039 ($\times1.56$) |
>
> Moreover, there remains room for engineering optimizations to further improve speed, such as speculative decoding [1, 2] based on the original OLLM. These strategies will help mitigate the additional computational cost while maintaining strong performance.
>
> [1] Leviathan et al. "Fast Inference from Transformers via Speculative Decoding." ICML 2023.
>
> [2] Yuan et al. "Speculative Contrastive Decoding." ACL 2024.

---

> > ### Author Response · Authors · 2025-11-20
> > **Response (2/2)**
> >
> > > **Q1:** What is the theoretical justification for using Jensen-Shannon divergence to measure disagreement between distributions? Were other metrics explored?
> >
> > Our adoption of JSD is also inspired by its application in existing work such as DoLa [1], which we have now properly cited in our revised manuscript. The use of Jensen-Shannon divergence (JSD) is theoretically motivated by its desirable properties: it is symmetric, always finite, and provides a stable measure of the difference between probability distributions.
> >
> > We explored several alternative metrics, including Wasserstein Distance (WD) and Cosine Distance (CD). The comparative results are presented below:
> >
> > | Setting  | MathVista | MMAU     | OmniBench |
> > | -------- | --------- | -------- | --------- |
> > | Baseline | 66.8      | 71.5     | 42.1      |
> > | JSD      | **70.2**  | 75.5     | **43.6**  |
> > | WD       | 69.7      | **75.7** | 43.3      |
> > | CD       | 66.3      | 72.3     | 42.5      |
> >
> > As shown in the experiments, JSD achieves the best performance on MathVista and OmniBench, while WD slightly outperforms JSD on MMAU. CD, on the other hand, performs noticeably worse than both JSD and WD, offering only marginal improvements over the baseline. Therefore, based on these experimental results and prior work [1], we chose JSD as our metric for measuring logits disagreement.
> >
> > [1] Chuang et al. "DoLa: Decoding by Contrasting Layers Improves Factuality in Large Language Models." ICLR 2024.
> >
> > > **Q2:** How does the method perform when the OLLM and LRM are from different model families or have significant architectural differences?
> >
> > In fact, as long as the OLLM and LRM share the same vocabulary, ThinkOmni can operate effectively, regardless of their underlying model families or architectures.  The corresponding experimental results are shown below:
> >
> > | Setting                                      | MathVista | MMAU | OmniBench |
> > | -------------------------------------------- | --------- | ---- | --------- |
> > | Qwen2.5-Omni-7B                              | 66.8      | 71.5 | 42.1      |
> > | +DeepSeek-distilled-7B (different families)  | 68.8      | 73.8 | 43.2      |
> > | +Qwen3-MoE 30B-A3B (different architectures) | 72.6      | 75.1 | 43.8      |
> >
> >
> > However, in cases where the OLLM and LRM do not share the same vocabulary, our current approach is unable to address the issue of mixing logits due to vocabulary mismatch. This limitation is also present in several other guidance decoding methods. Importantly, this challenge is widely recognized as a key research problem in the field of LLM ensembles [1,2]. As it involves broader and more complex topics that require further exploration, we regard it as an important direction for future work.
> >
> > [1] Bridging the Gap between Different Vocabularies for LLM Ensemble (Xu et al., NAACL 2024)
> >
> > [2] Ensemble Learning for Heterogeneous Large Language Models with Deep Parallel Collaboration (Huang et al., NeurIPS 2024)
> >
> > > **Q3:** How does ThinkOmni handle cases where the LRM provides incorrect reasoning? Is there a risk of propagating textual reasoning errors into omni-modal outputs?
> >
> > It is expected that the LRM may occasionally make mistakes, as our method is designed to leverage the existing reasoning capabilities of the LRM rather than to improve them directly. When conflicts arise between reasoning and perception, ThinkOmni dynamically adjusts the degree of logits mixing based on the relative importance of reasoning and perceptual information, thereby mitigating the impact of erroneous reasoning.
> >
> > We have added a failure case study in `Section 4.5`(see `Fig. 10(a)`, or access directly via [**this link**](https://anonymous.4open.science/r/ThinkOmni-8C48/Failure%20Case%20Study.png)). In this example, although ThinkOmni accurately perceives the image, incorrect reasoning by the LRM results in an incorrect final answer. Ideally, using a larger LRM would enhance overall performance, as larger models generally have stronger logical reasoning and reflective abilities and are less prone to errors.

---

### Official Review · Reviewer_Xh2x · 2025-11-03

**Soundness:** 3
**Presentation:** 4
**Contribution:** 3
**Rating:** 6
**Confidence:** 4

**Summary:**

The paper introduces THINKOMNI, a training-free framework designed to enhance the complex reasoning capabilities of Omni-modal Large Language Models (OLLMs). The authors identify that while OLLMs are good at perceiving diverse modalities (audio, video, image), they often lack the deep reasoning abilities found in recent text-only Large Reasoning Models (LRMs). To bridge this gap without expensive training or data annotation, the authors propose a decoding-time strategy with two main components:
- LRM-as-a-Guide: Leverages off-the-shelf text-only LRMs to guide the OLLM's token generation process via logit fusion.
- Stepwise Contrastive Scaling: A dynamic mechanism that adjusts the guidance weights at each decoding step based on the Jensen-Shannon divergence between different model distributions. This adaptively balances signals from perception (Omni-modal inputs) and reasoning (textual inputs).

Experiments across six multi-modal reasoning benchmarks (including Math Vista, MMAU, and OmniBench) demonstrate that THINKOMNI consistently improves the performance of base OLLMs (like Qwen2.5-Omni), sometimes matching or exceeding models trained with Reinforcement Fine-Tuning (RFT)

**Strengths:**

- The idea of using a text-only reasoning model to guide an omni-modal model during inference is a clever approach to addressing the scarcity of high-quality omni-modal reasoning data. While guidance decoding exists, applying it specifically to lift reasoning capabilities from one modality to many is a novel application.

- The proposed Stepwise Contrastive Scaling is a technically sound contribution. By using Jensen-Shannon divergence to measure the "disagreement" or unique contribution of reasoning ($D_R$) versus perception ($D_P$) at each step, the model avoids manual hyperparameter tuning and adapts to different task types (e.g., audio perception vs. math reasoning) on the fly.

- The framework is entirely training-free, avoiding the high computational costs and data requirements of Supervised Fine-Tuning (SFT) or RFT. This makes it practically valuable for rapidly deploying improved reasoning in existing omni-modal systems.

**Weaknesses:**

- (minor, ack'ed by the authors) The framework relies on logit fusion, which strictly requires the OLLM and the LRM to share the exact same tokenizer vocabulary. This is a major limitation to the claim of using "off-the-shelf" LRMs, as it restricts acceptable pairings to models within the same family (e.g., Qwen-based OLLMs with Qwen-based LRMs).

- Since the LRM never sees the omni-modal input and only sees the text trace, there is a theoretical risk that in highly visual/auditory tasks where the text prefix is ambiguous, the LRM might confidently "reason" in a wrong direction based only on text priors. It would be great to study such tricky cases / failure modes.

- While the method beats standard baselines, it would be useful to know if the ~3x compute cost at inference time would be better spent just using a much larger standard OLLM if one is available, rather than guiding a smaller one.

**Questions:**

See weaknesses.

---

> ### Author Response · Authors · 2025-11-20
>
> Thank you for your thorough review of our paper and for recognizing the novelty and practical value of our proposed method. We hope our responses below will address your concerns.
>
> >**W1:** The framework relies on logit fusion ... a major limitation to the claim of using "off-the-shelf" LRMs, as it restricts acceptable pairings to models within the same family.
>
> Our approach falls under the category of guidance decoding, where vocabulary consistency is a common limitation shared by such methods. Nevertheless, this limitation is not insurmountable. Recent studies in LLM ensembles have proposed effective strategies to fuse logits from models with different vocabularies [1, 2]. We agree that ensemble learning across modalities (e.g., combining VLMs and LLMs) is a promising direction and merits detailed future exploration.
>
> In the present work, we have focused our analysis on open-source models from the same family to ensure a controlled experimental setup. We sincerely appreciate your suggestion and will carefully consider exploring cross-modality ensemble methods in our future research.
>
> [1] Xu et al. "Bridging the Gap between Different Vocabularies for LLM Ensemble." NAACL 2024.
>
> [2] Huang et al. "Ensemble Learning for Heterogeneous Large Language Models with Deep Parallel Collaboration." NeurIPS 2024.
>
> > **W2:** Since the LRM ... the LRM might confidently "reason" in a wrong direction based only on text priors. It would be great to study such tricky cases / failure modes.
>
> We have carefully analyzed error patterns of our models and included a failure case analysis in the latest version of our paper. Specifically, we discuss two typical failure cases arising from the LRM’s inability to access the full multi-modal input. (See `Fig. 10`, or access directly via [**this_link**](https://anonymous.4open.science/r/ThinkOmni-8C48/Failure%20Case%20Study.png))
>
> - **Case 1：** `Fig. 10(a)` presents an example where the OLLM perceives the content correctly, but the LRM introduces incorrect guesses and reasoning. In this instance, the model recognizes that the beaker’s capacity is 600ml and that the highest visible marking is 400ml, leading to the erroneous inference that “the image only shows part of the beaker.”
>
> - **Case 2：** `Fig. 10(b)` demonstrates a scenario where the OLLM fails to accurately discern details in the input audio, resulting in incorrect downstream reasoning.
>
> > **W3:** While the method beats standard baselines, ... a much larger standard OLLM if one is available, rather than guiding a smaller one.
>
> Thank you for your insightful comments. While our method does involve both an OLLM and an LRM, leading to higher resource consumption than a single OLLM, this design represents a deliberate trade-off. It achieves significant performance improvements that would otherwise demand deploying much larger models, which entails even greater inference and training costs and often requires extensive, proprietary training data.
>
> Instead of directly competing with larger OLLMs, our method functions as a framework that leverages the strengths of high-performing LLM to maximize scalability. This is especially valuable in specialized domains where large, domain-specific models are either unavailable or lag behind mainstream ones. In such scenarios, our framework acts as a vital bridge, connecting state-of-the-art general-purpose LLMs with expert models for specific modalities. This facilitates rapid adaptation to new tasks and domains while circumventing the high costs associated with training dedicated large models from scratch.

---

### Author Response · Authors · 2025-11-26

Dear Reviewers,

Thank you all for giving us positive scores at the beginning of the discussion stage. We sincerely appreciate the constructive feedback and concerns raised by each of you, which have greatly helped us reflect on and improve our manuscript, especially in terms of algorithmic efficiency, failure case analysis, and general reasoning ability.

We have addressed your concerns in our previous response and revised the manuscript accordingly, with all changes highlighted in $\textrm{\color{blue}blue}$. As the discussion period is coming to an end with **one week** remaining, we kindly ask whether your concerns have been satisfactorily resolved. We are more than willing to provide further clarification if needed.

We look forward to your responses！

Best regards,

The Authors of Submission 8756

---

### Author Response · Authors · 2025-12-01
**Summary of Reviewer-Author Discussions and Key Updates**

Dear PCs, SACs, ACs, and Reviewers,

Thank you very much for your valuable contributions to our work. To assist the AC in the decision-making process, we provide below a summary of the key points from the reviews and how we have addressed the reviewers' concerns during the discussion period.

### **Strength**

Overall, we are grateful that the reviewers gave this paper a positive evaluation (all initial ratings are 6) and recognized the value of our training-free framework. Specifically:

*   **Novelty and Practicality.** The proposed ThinkOmni framework effectively enhances OLLMs using off-the-shelf LRMs without expensive training. All four reviewers recognized this point (**Xh2x**: Strength 1, 3; **myT4**: Strength 3, 4; **kNcB**: Strength 1; **2VXa**: Strength 1).
*   **Technical Soundness.** The Stepwise Contrastive Scaling mechanism is technically sound and effectively balances perception and reasoning signals. Three reviewers explicitly highlighted this (**Xh2x**: Strength 2; **myT4**: Strength 1; **kNcB**: Strength 1).
*   **Solid Performance.** The method demonstrates consistent and significant improvements across various benchmarks. All four reviewers recognized this point (**Xh2x**: Summary; **myT4**: Strength 2; **kNcB**: Strength 2; **2VXa**: Strength 3).

### **Concerns and Our Addressing**

During the discussion period, we actively addressed the reviewers' concerns with additional experiments and clarifications. Specifically:

**1. Inference Efficiency and Cost**
**(Xh2x: Weakness 3; myT4: Weakness 2; 2VXa: Weakness 1)**
Reviewers expressed concern that running both OLLM and LRM incurs computational overhead.

* **Our Addressing:** We clarified that our method avoids the massive cost of training large models. Crucially, we conducted **new experiments on parallel deployment**, showing that parallelizing the forward passes significantly reduces generation time (**0.072s $\to$ 0.039s**), making it competitive for real-time deployment.

**2. Error Propagation and Failure Modes**
**(Xh2x: Weakness 2; myT4: Question 3; 2VXa: Weakness 2)**
Reviewers asked about cases where the LRM might hallucinate or conflict with perception.

*   **Our Addressing:** We added a **Failure Case Analysis (Fig. 10)** in the revised paper. We explained how our Stepwise Contrastive Scaling (using JSD) naturally mitigates this by reducing the influence of LRM during perception-heavy steps. Reviewer **2VXa** explicitly acknowledged our reply regarding this mechanism.

**3. Domain Generalization (Beyond Math)**
**(kNcB: Weakness 1; 2VXa: Question 7)**
Reviewers suggested expanding benchmarks beyond math to include perception or common-sense reasoning.

*   **Our Addressing:** We added experiments on **MME, MMMU, and MMMU-Pro**. Results show that ThinkOmni achieves consistent improvements in common-sense reasoning tasks, demonstrating its ability to generalize beyond mathematical domains.

**4. Methodological Clarifications**

* **(myT4: Question 1)** Questioned the choice of JSD metric.

  **Our Addressing:** We provided comparative results showing **JSD outperforms Wasserstein Distance and Cosine Distance**, justifying our design choice.

* **(kNcB: Weakness 3)** Comparison with ProxyTuning.

  **Our Addressing:** We clarified that ThinkOmni outperforms ProxyTuning by >1% on key benchmarks and offers better scalability without domain-specific fine-tuning.

* **(kNcB: Weakness 2)** Confusion regarding Fig. 4 notation.

  **Our Addressing:** We revised Fig. 4 to clarify that the notation refers to the standard decoding process, not multi-turn dialogue.

### **Recognition of Our Revision**

Following our responses and revisions, **Reviewer 2VXa** explicitly acknowledged our efforts and **increased the Soundness score (3 $\to$ 4) and Confidence (4 $\to$ 5)**. We believe we have also satisfactorily addressed the questions from Reviewers **Xh2x**, **myT4**, and **kNcB**.

We hope this summary assists the AC in their final assessment. We once again thank the reviewers for their constructive feedback, which has significantly improved our paper.

Sincerely,

The Authors of Submission 8756

---

### Meta-Review · Area_Chair_raDs · 2026-01-06

**Summary:**

The paper proposes ThinkOmni, a training-free framework designed to enhance the reasoning capabilities of Omni-modal Large Language Models (OLLMs) by leveraging off-the-shelf Large Reasoning Models (LRMs) via guidance decoding. The reviewers generally acknowledged the novelty of the proposed "Stepwise Contrastive Scaling" mechanism and the practical appeal of a training-free approach, resulting in uniform initial ratings of 6.

1.  Effective Resolution of Efficiency Concerns: A primary concern raised by reviewers (Xh2x, myT4, 2VXa) was the inference overhead of running two models. The authors provided a compelling rebuttal by demonstrating new experiments showed that parallelizing forward passes significantly reduces latency (from 0.072s to 0.039s), making the approach competitive for deployment.

2.  Generalization Capabilities: In response to requests for broader evaluation beyond mathematical tasks (Reviewers kNcB, 2VXa), the authors expanded their experiments to include MME, MMMU, and MMMU-Pro. These additional results demonstrated consistent improvements in common-sense reasoning and perception-heavy tasks, partially proving the framework’s generalizability beyond the initial math-focused scope.

3.  Clarification on Failure Modes: The authors transparently addressed concerns regarding error propagation (i.e., the LRM hallucinating based on text only). They added a detailed failure case analysis (Fig. 10) and clarified how the contrastive scaling mechanism naturally down-weights the LRM's influence during perception steps, thereby mitigating the risk of hallucinations interfering with visual grounding.

The paper presents a plug-and-play solution for enhancing multimodal reasoning. The authors effectively addressed the reviewers' concerns regarding efficiency and generalization with new data and clarifications.

**Reviewer Concerns:**

While the proposed method of "Lifting" a smaller Omni-modal model using a larger Reasoning model (LRM) is intuitively appealing, several critical issues regarding practicality, experimental validation of claims, and baselines remain unresolved.

 1. Inference Overhead and Practicality

A major concern raised by multiple reviewers (reviewer Xh2x, myT4, 2VXa) was the computational cost of running two models simultaneously. In the rebuttal, the authors argued that parallelization strategies could reduce inference latency (from 0.072s to 0.039s).

While parallelization addresses latency, it does not solve the fundamental resource constraint. Running a large "Guide" model alongside the target OLLM significantly increases memory (VRAM) requirements. This severely limits the practical deployment of the framework. If a user possesses the hardware resources to run both models in parallel, the trade-off analysis of why they should not simply deploy a single, larger, superior OLLM is missing. While the efficiency argument remains less unconvincing, Acs chose not to use it against the acceptance.


2. Overclaiming on Performance and Generalization in the rebuttal.

The authors made several strong claims in their rebuttal and manuscript that are not adequately supported by the experimental evidence:
(1) "...achieves significant performance improvements that would otherwise demand deploying much larger models, which entails even greater inference and training costs..."
    This is an overclaim. The paper lacks a direct comparison with these "much larger models" to substantiate that ThinkOmni actually matches their performance. Without this upper-bound baseline, the claim is speculative. **I must point out that ThinkOmni uses the Qwen3 series as the LRM and the Qwen2.5-Omni as the OLLM, but the authors did not provide comparative experiments between ThinkOmni and stronger sota model Qwen3-Omni to support their claims in the main paper.**
(2) "vital bridge, connecting state-of-the-art general-purpose LLMs with expert models for specific modalities."
    This is also overclaimed. There is insufficient evidence to prove this works robustly as a general "bridge" between diverse, heterogeneous expert models and general LLMs, particularly given the logit-fusion constraints (reviewer Xh2x).

Since these over claimed sentences are not presented in the paper, the ACs just pointed them out here.


3. Reliability of the guidance

The framework relies on the reasoning capabilities of the LRM. However, the thinking process of such models may not be infallible. For example, even a model of Qwen3-Thinking-30B's size often exhibits hallucinations in long-context responses.

**Reviewer Scores:**

All reviewers can maintain their original scores, tending to the acceptance.

---

### Decision · Program_Chairs · 2026-01-26

Accept (Poster)